# The Pattern of Progression to First-Line Treatment with Dabrafenib and Trametinib in Patients with Unresectable or Metastatic, BRAF-Mutated, Cutaneous Melanoma: Results of the Observational T-WIN Study

**DOI:** 10.3390/cancers15071980

**Published:** 2023-03-26

**Authors:** Michele Del Vecchio, Vanna Chiarion Sileni, Pietro Quaglino, Gaetana Rinaldi, Alessandro Minisini, Teresa Troiani, Francesca Consoli, Andrea Sponghini, Maria Banzi, Maria Francesca Morelli, Dario Palleschi, Ernesto Rossi, Riccardo Marconcini, Roberta Depenni, Fabrizio Carnevale-Schianca, Ilaria Marcon, Paola Queirolo

**Affiliations:** 1Fondazione IRCCS Istituto Nazionale dei Tumori, 20133 Milan, Italy; 2Veneto Institute of Oncology IOV-IRCCS, 35128 Padova, Italy; 3Dermatology Clinic, Department of Medical Sciences, University of Turin, 10126 Turin, Italy; 4Aoup Paolo Giaccone Palermo, 90127 Palermo, Italy; 5Azienda Sanitaria Universitaria Del Friuli Centrale, 33100 Udine, Italy; 6Unit of Dermatology, University of Campania Luigi Vanvitelli, 80123 Naples, Italy; 7Medical Oncology, ASST Spedali Civili, 25123 Brescia, Italy; 8AOU Maggiore della Carità, DIMET, Università del Piemonte Orientale, 28100 Novara, Italy; 9Presidio Ospedaliero Arcispedale S. Maria Nuova AUSL di Reggio Emilia-IRCCS, 42123 Reggio Emilia, Italy; 10Istituto Dermopatico dell’Immacolata IDI-IRCCS, 00167 Rome, Italy; 11Presidio Ospedaliero S. Maria di Ca’ Foncello Azienda ULSS 2, Marca Trevigiana, 31100 Treviso, Italy; 12Fondazione Policlinico Universitario Agostino Gemelli-IRCCS, 00168 Rome, Italy; 13Presidio Ospedaliero S. Chiara, Azienda Ospedaliero Universitaria Pisana, 56100 Pisa, Italy; 14Department of Oncology and Hematology, University Hospital of Modena and Reggio Emilia, 41121 Modena, Italy; 15Candiolo Cancer Institute, FPO-IRCCS, 10060 Candiolo, Italy; 16Novartis Farma S.p.A, 20154 Milan, Italy; 17European Institute of Oncology-IRCCS, 20141 Milan, Italy

**Keywords:** metastatic melanoma, unresectable melanoma, dabrafenib, trametinib, lactate dehydrogenase, BRAF V600 mutation

## Abstract

**Simple Summary:**

This study collected data from patients with cutaneous melanoma with mutation in the B-RAF gene who were treated with the combination of dabrafenib and trametinib in 34 Italian hospitals in every-day clinical practice. The aim was to understand how these drugs modify the progression pattern of the tumor in a heterogeneous population of patients who had either a limited (104 patients) or a bulky disease (97 patients). The results showed that fewer patients had lesions at the skin and more patients had lesions in other sites rather than the skin and lymph nodes when the disease progressed compared to baseline, and the proportion of patients with at least three involved organs at the progression increased in both cohorts. The survival and the time without progression were similar to previously published data and no new adverse reactions were reported. Therefore, dabrafenib and trametinib demonstrated to be effective and safe in heterogenous patients.

**Abstract:**

In patients with B-RAF-mutated cutaneous melanoma, targeted therapies are the treatment of choice to achieve a rapid response. In this multicentric, prospective, observational study, patients with B-RAF-mutated cutaneous melanoma who were treated with dabrafenib and trametinib were categorized in two cohorts (cohort A: limited disease (n = 104) and cohort B: bulky disease (n = 97)) according to lactate dehydrogenase levels. The primary endpoint was the progression pattern; the secondary endpoints were overall survival (OS), progression-free survival (PFS), and safety data. From baseline to time of progression, there was a progression from nodal to other sites of disease in cohort A and from skin and nodal to other sites in cohort B. In both the cohorts, the number of involved organs and metastases at each location decreased. The median OS was 32.4 months (95% CI: 20.1 months (not estimable)) for cohort A, and 10.5 months (95% CI: 8.3–14.4 months) for cohort B; median PFS was 12.4 months (95% CI: 10.9–17.0 months) for cohort A, and 8.1 months (95% CI: 6.3–9.4 months) for cohort B. No new safety signals were reported. This study describes the patterns of first-line treatment progression with dabrafenib and trametinib in Italian clinical practice. The effectiveness and safety data were consistent with previous trials and extended to a real-world heterogeneous population.

## 1. Introduction

Worldwide cutaneous melanoma accounts for 1.7% of cancer diagnoses with approximately 325,000 new cases in 2020; it is the most aggressive form of skin cancer, causing almost 57,000 disease-related deaths each year. According to SEER, the 5-year survival rate for those first diagnosed with stage I–II disease is 99.4%, decreasing to 68.0% for stage III, and 29.8% for stage IV. Only 4% of diagnoses are made in stage IV, while 83% of diagnoses are made in stages I–II [1,2].

Most cutaneous melanomas show an aberrant activation of the mitogen-activated protein kinase (MAPK) pathway and mutations in proteins along the RAS-RAF-MEK-ERK pathway: 50% B-RAF-mutated and 15% NRAS-mutated [3]. This pathway is the target of the combination of dabrafenib and trametinib, which in 2014, was approved for the treatment of cutaneous melanoma harboring V600E or V600K-BRAF mutation by the Food and Drug Administration.

Dabrafenib blocks MAPK signaling in patients in the presence of V600E B-RAF mutation and it improves the overall survival (OS) and progression-free survival (PFS) in patients with previously untreated melanoma. Trametinib mediates the blockade of MAPK kinase (MEK), which is downstream of B-RAF in the MAPK pathway. However, after 6–8 months of treatment with monotherapy of both dabrafenib and trametinib, resistance to B-RAF inhibition rapidly emerges [4].

Combining these two agents is an attempt to delay or overcome the resistance [5].

In phase 3 trials, the combination of dabrafenib (150 mg orally twice daily) and trametinib (2 mg orally once daily) improved the PFS rate in previously untreated patients who had metastatic melanoma with BRAF V600E or V600K mutations compared to dabrafenib alone and placebo [6], and prolonged OS compared to vemurafenib, without increasing the overall toxicity [7]. In these phase 3 trials, the dabrafenib plus trametinib combination was investigated in a population that included patients with various prognostic aspects; only the lactate dehydrogenase (LDH) levels were distinguished through stratification.

The analysis of the clinicopathological features of those experiencing long-term benefits and those who progress may contribute to the right selection of patients for the combination. Furthermore, as the registration trials did not thoroughly study the possible causes of treatment progression and the impact of dabrafenib and trametinib on the clinical outcomes of subsequent treatment lines, the evidence on this topic is scarce.

This observational study aimed to follow cutaneous melanoma patients with either a limited or bulky disease burden [8] treated with combined dabrafenib and trametinib in an Italian clinical practice according to the local label. Data about patterns of first-line treatment progression with this combination and its influence on second-line treatment outcomes were collected prospectively from the initial visit until progression during second-line treatment.

## 2. Material and Methods

### 2.1. Study Design

This was a multicentric, prospective, observational study, involving patients with B-RAF-mutated cutaneous melanoma who were treated with dabrafenib and trametinib. The treatment was independent to the participation in the study and patients were managed at the discretion of the physician according to the label and local recommendations, as per routine clinical practice.

### 2.2. Setting

The study was conducted at 34 hospitals located in Italy and it lasted approximately 4 years. Each patient was observed for a maximum of 12 months after starting the second-line treatment or until the second progression (whichever came first). The study was approved by the ethical committee as per local regulations (Ethics committee of the National Cancer Institute of Milan, Coordinator Center, approval number: INT 93-17; the study was approved by all ethics committees of the involved hospitals), and was conducted per the World Medical Association Declaration of Helsinki. Patients were required to sign an informed consent before data collection. All the data referring to the patients were published anonymously, without any details allowing their re-identification.

### 2.3. Participants

To be eligible for the study, patients had to be ≥ 18 years old, have a confirmed diagnosis of BRAF V600E/K or other BRAF mutant advanced or metastatic melanoma, be naïve to treatment for advanced/metastatic disease without having received prior systemic anti-cancer treatment (chemotherapy, immunotherapy, biological therapy, vaccine therapy, or investigational treatment) for Stage IIIC (unresectable) or Stage IV (metastatic) melanoma (treatments in adjuvant setting were admitted), and be assigned to first-line treatment with labeled use of dabrafenib and trametinib combination.

Patients were divided into two cohorts according to the presence of limited disease burden (lactate dehydrogenase LDH ≤ upper limit of normal ULN, cohort A) or bulky disease (LDH > ULN, cohort B), and data from up to 100 patients were collected in each group.

Patients were analyzed for patterns of first-line treatment progression at the time of progression (core phase). Those patients who discontinued treatment due to progression and who were assigned to second-line treatment entered the extension phase of the study and were followed up for 12 months after starting the second-line treatment or until second progression (whichever came first), in order to investigate how dabrafenib plus trametinib first-line treatment could influence subsequent treatment outcomes (Appendix A).

### 2.4. Variables

In the first approved version of the protocol, the primary endpoint was the response/progression pattern at the expected median PFS time (17.5 months for cohort A and 5.5 months for cohort B). Information on site of progression (local, nodal, or distant) versus sites of disease at baseline, number of organs involved in progression versus baseline, number of metastases for each organ versus baseline, median time to development of new metastasis from treatment initiation, and Eastern Cooperative Oncology Group Performance status (ECOG PS) at time of progression versus baseline were collected. However, since some inconsistencies between the endpoint of the study and the statistical methods adopted for evaluation emerged, both the primary endpoint and the methods were amended. The primary endpoint was evaluated at the time of progression with a 3-month window and not at prespecified timelines, and only the progression pattern (and not response/progression pattern) was analyzed.

The secondary endpoints were the correlation between the progression pattern of first-line treatment with the investigator-reported response to second-line systemic treatment, OS, PFS, and best response to first-line and second-line treatment. The median duration of dabrafenib and trametinib and second-line treatment, dose modification of both treatments due to adverse events, the incidence of serious and non-serious adverse events, and quality of life were also described.

### 2.5. Measurement

Disease progression was documented by the treating physician(s), as per routine clinical practice.

Time to new metastasis was defined as the time from initiation of dabrafenib and trametinib treatment to the date of assessment of new metastasis or death due to any cause. For patients who neither developed new metastasis nor died, time was censored at the date of their last available assessment.

To assess changes in the quality of life, the 5-level EuroQol 5 dimensions (EQ-5D-5L) questionnaire, which describes general health, was used, while the overall work impairment due to health was assessed with the Work Productivity and Activity Impairment: General Health (WPAI-GH) questionnaire.

### 2.6. Study Size

No formal statistical power calculations to determine sample size were performed for this study; the predefined sample size of 100 patients included in each cohort was based on the prevalence and access rate of eligible patients to the investigational sites.

### 2.7. Statistical Methods

Continuous data were summarized by number (n), mean, standard deviation (SD), median, and first and third quartiles. Categorical data were presented as the number and percentage of patients in each category. Given the observational nature of the study, all statistical analyses had descriptive purposes. Unless stated otherwise, a 2-sided alpha level of 0.05 was considered. No alpha-level adjustment was carried out for the primary and secondary outcome variables. Analysis datasets and statistical outputs were produced using SAS^®^ Version 9.4 (SAS Institute Inc., Cary, NC, USA).

## 3. Results

### 3.1. Participants and Demographic/Baseline Characteristics

From November 2017 to May 2019, 205 patients were enrolled in the study: 3 did not complete the screening phase, 1 completed the screening but did not receive the study medication, and 201 were treated and, therefore, included in the analysis. A total of 104 patients were included in cohort A and 97 in cohort B; all patients finished the core phase, and 89 patients entered the extension phase of the study (Appendix A).

The baseline characteristics are summarized in Table 1. The mean (SD) age of patients in cohort A was 60.6 (15.39) years and 62.4 (13.63) years in cohort B, with most patients being 18 to <65 years in both cohorts. The majority of patients were male, and almost all were Caucasian. ECOG PS was 0 in 82.7% of cohort A and 57.7% of cohort B; 77.9% of patients in cohort A and 83.5% of patients in cohort B had at least one medical condition.

A total of 94.2% of patients in cohort A and 84.5% of patients in cohort B had prior antineoplastic surgery, including excision/removal/ablation and biopsy; residual disease was present in 14.4% of patients in cohort A and 15.5% of patients in cohort B. A total of 5.8% of patients in cohort A and 8.2% of patients in cohort B had prior antineoplastic radiotherapy, and the brain was the most frequent location, followed by bone, and cervical lymph nodes. A total of 9.6% of patients in cohort A and 12.4% of patients in cohort B had at least one prior antineoplastic regimen of medication in adjuvant setting (mainly interferon, since in Italy pembrolizumab, nivolumab, dabrafenib, and trametinib became available in 2019).

At the baseline visit, 14 (13.5%) patients in cohort A and 37 (38.1%) patients in cohort B had metastatic brain lesions.

### 3.2. Primary Endpoints

#### 3.2.1. Site of Progression (Skin, Nodal, or Other) versus Sites of Disease at Baseline

The pattern of first-line treatment progression is presented in Figure 1.

For cohort A, most patients (85.6%) at baseline had a site of disease other than the skin and lymph nodes, followed by nodal (51.9%), and skin (15.4%). The proportion of patients with skin as the site of disease decreased from baseline to time of progression, whereas the proportion of patients with nodal and other sites of disease increased from baseline to time of progression. The proportions of patients with only skin as the site of disease and with only other sites of disease at baseline were similar to that at the time of progression, whereas the number of patients with a combination of sites of disease increased.

For cohort B, the majority (91.8%) of patients at baseline had other sites of disease, followed by nodal (57.7%), and skin (17.5%). From baseline to the time of progression, the proportion of patients with skin and nodal sites of disease decreased, whereas the proportion of patients with other sites of disease increased. The proportion of patients with only skin as the site of disease at baseline was similar to that at the time of progression, whereas patients with the only nodal site and with a combination of sites decreased. The most common shift was from skin to other and from nodal to other, followed by from other to nodal.

#### 3.2.2. Number of Organs Involved in Progression versus Baseline

In cohort A, the proportions of patients with one and two organs involved decreased from the baseline to the time of progression, whereas the proportions of patients with three and >three organs involved increased. The number of organs involved increased from one to two in four patients and from three to >three in two patients.

In cohort B, patients with one organ involved decreased, patients with two organs involved were similar at baseline and at the time of progression, whereas patients with three and >three organs involved increased. The number of organs involved increased from two to three in three patients and from three to >three in six patients.

#### 3.2.3. Number of Metastases for Each Organ versus Baseline and Time to Develop a New Metastasis

The number of metastases for each organ at the time of progression was either unchanged from baseline or missing for most of the patients in both cohorts. The median time to develop subsequent new metastases was 19.0 months (95% CI: 12.5, 38.5 months) for cohort A and 13.0 months (95% CI: 9.9, 23.3 months) for cohort B.

#### 3.2.4. ECOG PS at Time of Progression vs. Baseline

For cohort A, data on ECOG PS at the time of progression were available for 24 patients. The proportion of patients with ECOG PS of zero decreased from baseline to time of progression (85.1% vs. 70.8%), whereas the proportions of patients with ECOG PS of one and two increased from baseline to time of progression (12.9% vs. 20.8%, and 2.0% vs. 8.3%, respectively).

For cohort B, data were available for 25 patients. The proportions of patients in cohort B with ECOG PS of zero and three decreased from baseline to time of progression (62.2% vs. 56.0%, and 2.2% vs. 0%, respectively), whereas the proportions of patients with ECOG PS of one and two increased (28.9% vs. 32.0%, and 6.7% vs. 12.0%, respectively).

### 3.3. Secondary Endpoints

#### 3.3.1. Clinical Benefit

The estimated median OS was 32.4 months (95% CI: 20.1 months, (not estimable)) for cohort A, and 10.5 months (95% CI: 8.3 months, 14.4 months) for cohort B (Figure 2a), whereas the median PFS was 12.4 months (95% CI: 10.9, 17.0 months) for cohort A, and 8.1 months (95% CI: 6.3, 9.4 months) for cohort B (Figure 2b). Appendix A summarizes the subgroup analyses for OS and PFS by metastatic site at baseline. We highlighted the results of OS and PFS in cohort B in patients with brain metastasis (OS = 8.3 months (95% CI 6.6–11.7) and PFS =6.9 months (95% CI 5.2–9.4)) (Figure 3a,b), and liver metastasis (OS = 6.8 months (95% CI 5.6–8.2) and PFS = 5.7 months (95% CI 4.2–6.6)).

In both cohorts, most patients had a partial response as the best response to first-line treatment. For cohort A, partial response was more frequently observed in 51.9% of patients, followed by a complete response (18.3%; 95% CI: 11.4%, 27.1%), stable disease (12.5%), and progressive disease (8.7%). For 8.7% of patients, the best response to first-line treatment was missing.

For cohort B, 39.2% of patients had a partial response, followed by stable disease (22.7%), progressive disease (20.6%), and complete response (10.3%; 95% CI: 5.1%, 18.1%). For 7.2% of patients, the best response to first-line treatment was missing.

#### 3.3.2. Impact of Progression Pattern during First-Line Treatment on Disease Progression in Patients Receiving Second-Line Treatment

Univariate and multivariate logistic regression analysis of the impact of progression patterns during first-line treatment on second-line tumor response gave few statistically significant results (Appendix A). Following forward selection, only the number of metastases (>three vs. one) had a statistically significant impact on the second-line tumor response. Patients with >three metastases had higher odds of disease progression in second-line treatment compared to those with one metastasis (OR: 3.82; 95% CI: 1.07, 13.63; *p*-value: 0.0393). ECOG PS at the time of the first progression had no statistically significant impact on the second-line tumor response.

### 3.4. Quality of Life

At baseline, the EQ-5D-5L health status scores were available for 92 patients in cohort A and 82 patients in cohort B; at 6 months, the health status scores were available for 60 patients in cohort A and 43 patients in cohort B. The EQ-5D-5L health status increased, i.e., it improved, from baseline to 6 months for both cohorts with a mean change from baseline of 2.31 (SD: 16.494; 95% CI: −2.0, 6.6) for cohort A and 8.33 (SD: 23.313; 95% CI: 1.1, 15.6) for cohort B. For cohort A, there was a mean increase in the EQ-5D-5L health status score from baseline to month 6 for patients with brain metastases (month 6, n = 8; 3.13; SD: 8.425; 95% CI: −3.9, 10.2), but there was a mean decrease for patients with liver metastases (month 6 n = 15; −0.53; SD: 9.062; 95% CI: −5.6, 4.5). For cohort B, the mean increase in the EQ-5D-5L health status score from baseline to month 6 was greater for patients with brain metastases (month 6, n = 12; 15.42; SD: 26.668; 95% CI: −1.5, 32.4), than for patients with liver metastases (month 6, n = 12; 5.42; SD: 32.295; 95% CI: −15.1, 25.9).

At baseline, WPAI-GH percent activity impairment due to health was available for 88 patients in cohort A and 81 patients in cohort B. The baseline mean (SD) percent impairment was 34.55% (33.215%) for cohort A and 47.90% (35.204%) for cohort B.

At 6 months, percent activity impairment was available for 55 patients in cohort A and 40 patients in cohort B. The mean (SD) percent impairment was 31.64% (30.415%) for cohort A and 36.25% (32.081%) for cohort B at month 6. This resulted in a mean change from the baseline of 1.27% (SD: 33.503%; 95% CI: −7.8%, 10.3%) for cohort A and −3.75% (SD: 38.874%; 95% CI: −16.2%, 8.7%) for cohort B. For cohort A, the mean percent activity impairment due to health increased from baseline to month 6 for patients with brain metastasis (month 6, n = 8; 1.25%; SD: 22.952%; 95% CI: −17.9%, 20.4%), but decreased for patients with liver metastasis (month 6, n = 15; −3.33%; SD: 31.091%; 95% CI: −20.6%, 13.9%). For cohort B, the mean decrease in percent activity impairment from baseline to month 6 was greater for patients with liver metastasis (month 6, n = 12; −15.00%; SD: 31.479%; 95% CI: −35.0%, 5.0%), than for patients with brain metastasis (month 6, n = 12; −1.67%; SD: 34.859%; 95% CI: −23.8%, 20.5%).

### 3.5. Safety

A total of 96.2% of patients in cohort A and 92.8% in cohort B had at least one AE. Pyrexia was the most frequent adverse reaction, reported in 56.7% of patients in cohort A and 45.4% of patients in cohort B; asthenia, headache, and rash were described in more than 10% of patients in both cohorts. Other adverse reactions reported in at least 10% of patients in at least one cohort are summarized in Table 2. A total of 17.3% of patients in cohort A and 16.5% of patients in cohort B had at least one AE leading to study drug discontinuation, whereas 65.4% of patients in cohort A and 55.7% of patients in cohort B required dose adjustment or interruptions for AE.

Forty-three patients (41.3%) in cohort A and 70 patients (72.2%) in cohort B died during the study for disease progression.

No new safety signal emerged from the study.

## 4. Discussion

This study involved only Italian centers and described the use of dabrafenib and trametinib in an Italian real-world practice to treat patients with either limited or bulky BRAF V600E/K or other BRAF-activating mutation-positive cutaneous melanoma. In particular, this study was focused on prospectively examining the patterns of first-line treatment progression with dabrafenib plus trametinib combination.

From baseline to time of progression, in cohort A there was a general progression from nodal to other sites of disease, whereas in cohort B there was a shift of progression from skin and nodal sites to other sites. In both cohorts, the number of organs involved and the number of metastases at each tumor location decreased, whereas ECOG PS increased; the median time to develop new metastases from treatment initiation was longer for cohort A than for cohort B. This latter result was expected, as cohort A included patients with a better prognosis than cohort B.

These patterns of progression, disease kinetics (i.e., lactate dehydrogenase), and the disease burden, which decreased or did not change during the treatment with dabrafenib and trametinib, were associated with the clinical benefit in terms of OS, PFS, response rate, and maintenance of the quality of life.

These results mirrored those described in the pooled analysis of pivotal trials of Combi-v and Combi-d, where lactate dehydrogenase levels two times above the normal limit and more than three metastatic sites correlated to lower 1-year PFS and to 1-year OS rates [9].

Blocking the B-RAF-dependent pathway may affect the progression pattern of the disease and, therefore, the outcomes. Melanoma can metastasize to virtually any organ or tissue, including some sites that are rarely involved in other solid tumors [10]. The skin, subcutaneous tissue, and lymph nodes are the most common initial sites of distant metastases and they occurred in 42% to 59% of patients [11]. In approximately 25% of the patients, metastatic melanoma spreads at the visceral level, and the most common sites were the lung (18–36%), brain (12–20%), liver (14–20%), and bone (11–17%) [11]. Our study confirmed these observations, as patients most frequently showed metastases in other sites than the skin and lymph nodes.

The site of distant metastasis is an important independent predictor of survival in patients with metastatic disease. The median survival time of patients with melanoma metastasis to visceral sites other than the lung (M1C) was 7 months, while patients with lung metastases had a median survival time of 12 months. The survival time improved (18 months) if metastasis developed in non-visceral sites (i.e., skin, subcutaneous tissue, and distant lymph nodes). Therefore, evidence from this study indicates that patients with visceral metastases in the liver, brain, or bone, have a poor prognosis and limited response to treatments [12]. The analysis of individual patient data of Combi-d and Combi-v trials confirmed this observation, as the survival after progression was longer in patients with progression in baseline or new non-central nervous system (CNS) lesions (n = 205; median 10.0 months (95% CI 7.9–12.0)) than in those with new CNS lesions or concurrent progression in baseline and new lesions (n = 171; median 4.0 months (3.5–4.9)) [9]. The presence of liver metastasis diminished the immunotherapy efficacy [13], which was effective in a few patients harboring B-RAF mutations, as described in the Combi I trial [14].

Compared to clinical trials, where patients with brain metastasis are often excluded, real-world studies provide evidence on these patients. We obtained similar results in terms of OS and PFS in patients with brain metastasis and without brain metastasis in both cohorts. However, we observed a difference in OS between two cohorts in patients with liver or brain metastasis at baseline. In patients with limited disease and brain or skin metastasis, OS was not reached, while in patients with bulky disease and brain metastasis, OS was longer (8 months) than that reported in patients with liver metastasis (6.8 months). These results were consistent with the analyses of the retrospective study DESCRIBE III, which categorized patients according to three benefit groups based on the observed duration of treatment within the Nominal Patient Program/Individual Patient Program [15]. In this study, brain metastases were similar across all groups (short-term, 22.1%; long-term, 17.3%), while liver metastases were the highest in the short-term duration of the benefit group (short-term, 28.3%; long-term, 11.1%), thus corroborating the notion that the site of metastasis affects the clinical response to therapy [15]. In the retrospective study DESCRIBE II, patients who were naïve to B-RAF inhibitors were analyzed by the brain metastases’ status at treatment initiation [16]. In these patients the overall response rate was 61.3 in the presence of brain metastases and 71.0% in the absence; the median OS and PFS were 15.5 and 6.2 months, respectively, in patients with brain metastases, and 20.0 and 8.0 months, respectively, in patients without brain metastases [16]. The Italian experience of the Managed Access Program, as presented in the retrospective study DESCRIBE in Italy, confirmed the effectiveness and safety of dabrafenib and trametinib in unselected patients [17]. Furthermore, another single-center, retrospective analysis of 52 patients treated with B-RAF and MEK inhibitors for advanced melanoma over 12 months described disease progression in 59.6% of patients, of whom 70.9% had metastasis in the CNS, and the median time until a relapse was 8 months and the median survival time after progression was 2 months [18]. The progression in both new and pre-existing metastases was described in 52% of patients treated with B-RAF inhibitors (vemurafenib 82.2% and dabrafenib 17.8%), namely, exclusive extracranial progression occurred in 50.6% of the patients compared to both extra- and intracranial (29.4%) or sole cerebral progression (20%) [19]; the single-site progression and primary response to BRAF inhibitors were associated with an improved PFS [19]. Another retrospective real-world study reported an overall response rate of 39.8%, with an OS of 22.6 months and PFS of 9.2 months [20].

No new safety signals emerged from the analysis. The most common adverse reactions were pyrexia, asthenia, and rash, thus supporting previous evidence from clinical trials and the DESCRIBE studies that have demonstrated the good handling and tolerance of these drugs. Therefore, the treatment was safe and manageable in most patients.

Furthermore, our data highlighted that the treatment with dabrafenib and trametinib did not impair the quality of life and the ability to work and indeed, in both cohorts, the quality of life measured by the EQ-5D-5L health status improved during the treatment. What is noteworthy is that this improvement was also seen in patients with brain metastases, who are characterized by poor prognosis. In cohort B, an increase in the EQ-5D-5Lscore from baseline was observed even in patients with liver metastases, whereas this result was not achieved in cohort A. The treatment did not affect the ability to work, even in this case, which was measured by a validated questionnaire.

The study has some limitations. Due to the observational nature of the study, data were non-standardized since they were collected within the routine medical care framework where the medical practice may vary. For the primary objective, the data were limited, as less than half of each cohort had data at the time of progression compared to baseline, due to many patients being discontinued from the study for reasons other than disease progression; therefore, any trends should be interpreted with caution. Finally, as this study was terminated early, at a time when many patients were still being treated with the study drug, the full pattern of disease progression for these patients could not be assessed.

## 5. Conclusions

This study of approximately 200 patients with B-RAF-mutant cutaneous melanoma likely represents the patterns of first-line treatment progression with combined dabrafenib and trametinib treatment in an Italian clinical practice. The effectiveness, quality of life, and safety data reported here were consistent with those observed in the pivotal phase 3 clinical trials and were thus extended to a real-world heterogeneous patient population. This population included patients with either limited or bulky diseases, as well as patients with brain metastases and an ECOG PS > 1, who are commonly associated with poor prognosis and are excluded from clinical trials. Further studies on patterns of progression would contribute to better predictions of the outcomes of patients with metastatic cutaneous melanoma and their responses to the currently available first- and second-line treatments.

## Figures and Tables

**Figure 1 cancers-15-01980-f001:**
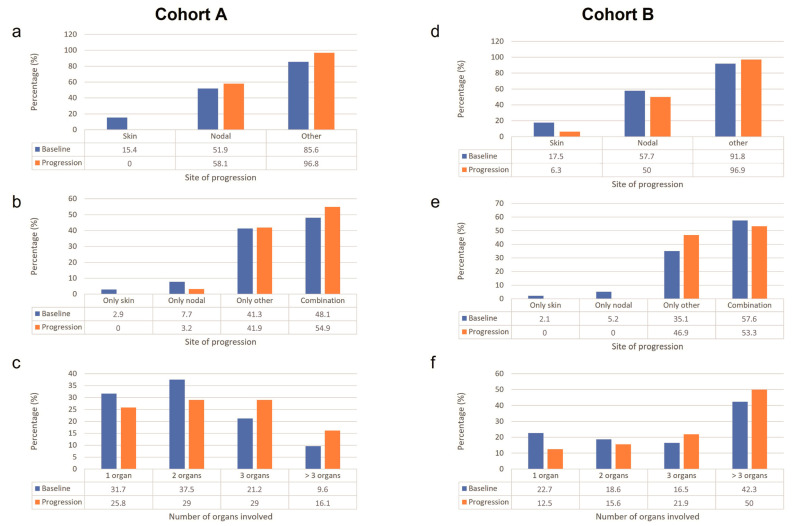
Disease progression. Patterns of progression in patients with limited disease (cohort A—(**a**–**c**)) or bulky disease (cohort B—(**d**–**f**)). Panels (**a**) and (**d**) describe the overall changes in the site of progression, showing an increase in the proportion of patients with lesions at lymph nodes and other sites at the time of progression. Panels (**b**) and (**e**) represent the proportion of patients who presented a lesion only in the skin, lymph nodes, other sites, or in a combination of sites; the most relevant change was observed in cohort B for the proportion of patients with lesions in other sites that increased at the time of progression. Panels (**c**) and (**f**) describe the proportion of patients according to the number of involved organs, showing an increase in the proportion of patients with at least 3 organs involved in both cohorts.

**Figure 2 cancers-15-01980-f002:**
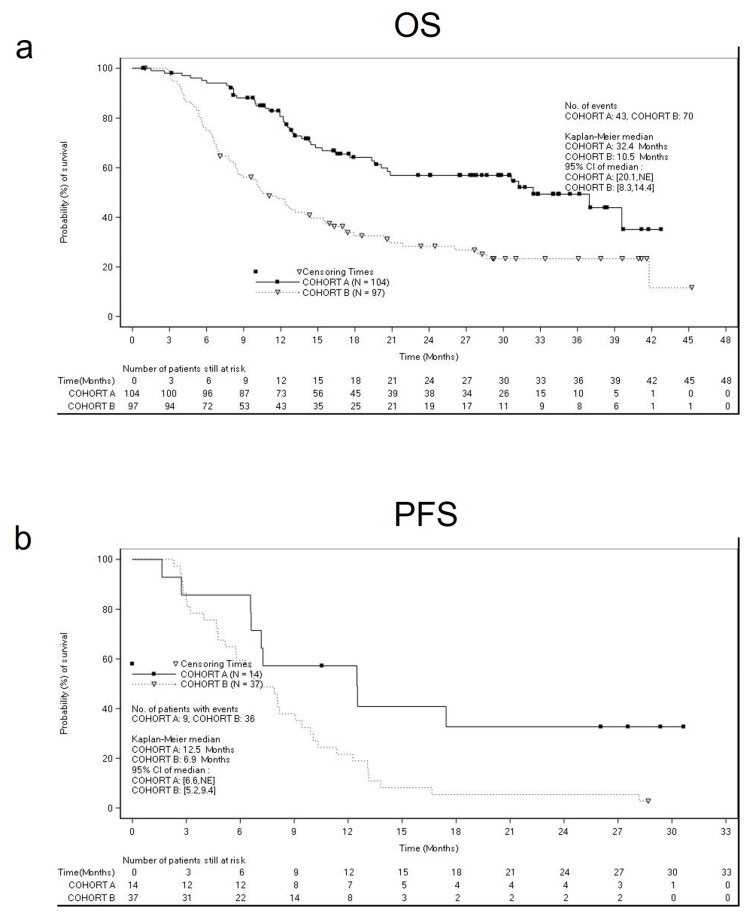
Kaplan–Meier estimates of overall survival (**a**) and progression-free survival (**b**) in the cohorts. Panel (**a**) shows the median overall survival in cohort A (32.4 months) and cohort B (10.5 months). Panel (**b**) shows the median progression-free survival in cohort A (12.5 months, with 9 events occurred) and cohort B (6.9 months, with 36 events occurred).

**Figure 3 cancers-15-01980-f003:**
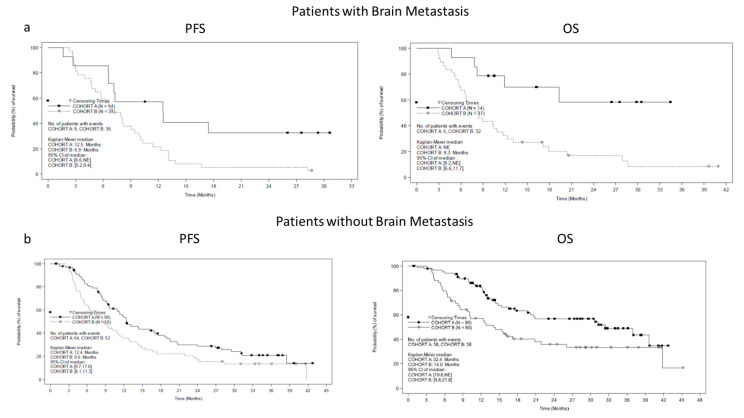
Progression-free survival and overall survival according to brain metastasis in cohorts A and B. PFS and OS in patient with brain metastasis are shown in panel (**a**). In the panel (**b**), PFS and OS in patient without brain metastasis are reported.

**Table 1 cancers-15-01980-t001:** Demographic and baseline characteristics.

	Cohort A (n = 104)	Cohort B (n = 97)
Sex, n (%)		
Male	65 (62.5)	62 (63.9)
Female	39 (37.5)	35 (36.1)
Age, mean (SD)	60.0 (15.4)	62.4 (13.6)
18 to <65 years	61 (58.7)	49 (50.5)
65 to ≤85 years	41 (39.4)	45 (46.4)
>85 years	2 (1.9)	3 (3.1)
Race, n (%)		
Caucasian	103 (99.0)	96 (99.0)
Unknown	1 (1.0)	1 (1.0)
Body Mass Index, mean (SD)	26.05 (4.13)	25.30 (4.16)
Missing	35	23
ECOG PS, n (%)		
0	86 (82.7)	56 (57.7)
1	13 (12.5)	26 (26.8)
2	2 (1.9)	6 (6.2)
3	0	2 (2.1)
Missing	3 (2.9)	7 (7.2)
Medical history/concomitant medical conditions (%)	77.9 *	83.5 *
Hypertension	30.8	38.1
Hypercholesterolemia	5.8	10.3
Benign prostatic hyperplasia	7.7	3.1
Diabetes mellitus	5.8	6.2
Appendicectomy	4.8	6.2
Prior antineoplastic therapy	95.2	84.5
Surgery	94.2	84.5
Radiotherapy	5.8	8.2
Adjuvant antineoplastic medication	9.6	12.4

* At least one medical condition.

**Table 2 cancers-15-01980-t002:** Adverse reactions. Data were reported as percentage.

	Cohort A	Cohort B
Pyrexia	56.7	45.4
Asthenia	23.1	25.8
Fatigue	14.4	9.3
Headache	14.4	13.4
Rash	13.5	10.3
Nausea	9.6	17.5
Vomiting	11.5	8.2
Diarrhea	9.6	13.4
Cough	10.6	6.2

## Data Availability

All data can be found in the text.

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
