# Peer review of "The Pattern of Progression to First-Line Treatment with Dabrafenib and Trametinib in Patients with Unresectable or Metastatic, BRAF-Mutated, Cutaneous Melanoma: Results of the Observational T-WIN Study"

_cancers, 2023, doi:10.3390/cancers15071980_

Round 1

Reviewer 1 Report (Previous Reviewer 1)

In response to the question about what proportion of patients with Stage IV disease had skin lesion at baseline and the suggestion that the authors may want to analyze the data by stratifying patients based on disease stage, the authors stated that  the efficacy analysis after stratifying was not done. They do not explain why it was not done!

Author Response

REV#1

Comments and Suggestions for Authors

In response to the question about what proportion of patients with Stage IV disease had skin lesion at baseline and the suggestion that the authors may want to analyse the data by stratifying patients based on disease stage, the authors stated that  the efficacy analysis after stratifying was not done. They do not explain why it was not done!

First, we apologize if our previous answer was not complete and, second, we thank the reviewer to point it out again, giving us the chance to explain it better. The reason because the  efficacy analysis after stratifying for stage III and IV was not done is that due the small sample number (11 patients in stage III not resectable) when they were separately analysed, no statistical or clinical meaning has been expected.

Reviewer 2 Report (Previous Reviewer 2)

Thank you for revised version of your ms. I acknowledge that you have not captured data on primary tumors and toxicity in the data collection.

Author Response

REV#2

Comments and Suggestions for Authors

Thank you for revised version of your ms. I acknowledge that you have not captured data on primary tumors and toxicity in the data collection.

We confirm that data on primary tumors were not collected.On the other hand, data on toxicity have been described in our manuscript [3.5. Safety; Table 2]. On this regard, we want to remember that cardiological toxicity, left ventricular ejection fraction decrease due to BRAF and MEKis, appears quite common but usually not severe, without impact on patient outcomes, as previously reported (Berger et al , 2020).  So, data on fraction ejection were not collected/analysed in our patients, even because no standardized protocol was assessed but only a recommendation to perform an ECG was advised to clinicians. 

Reviewer 3 Report (Previous Reviewer 3)

The authors did not answer all the notes, at least they should write something what they think:

The authors already reported an investigation entitled: Retrospective Chart Review of Dabrafenib Plus Trametinib in Patients with Metastatic BRAF V600-Mutant Melanoma Treated in theIndividual Patient Program (DESCRIBE Italy) by Massimo Aglietta et al in Target Oncol 2021 Nov;16(6):789-799. doi: 10.1007/s11523-021-00850-1.

This study is a prospective study with limitation as described by the authors „Due to the observational nature of the study, data were non-standardized since they were collected within the routine medical care framework where the medical practice may vary.

The other major limitation was : „this study was terminated early, at a time when many patients were still being treated with the study drug, the full pattern of disease progression for these patients could not be assessed.

My opinion is that it is very hard to follow the results as described. The clinical parameters of patients are missing. The descriptions of supplementary Figures are superficial and the titles of Figures are superficial.Other notes: Simple summary: It is not clear what are the new observation of this study.

Author Response

REV#3

Comments and Suggestions for Authors

The authors did not answer all the notes, at least they should write something what they think:

The authors already reported an investigation entitled: Retrospective Chart Review of Dabrafenib Plus Trametinib in Patients with Metastatic BRAF V600-Mutant Melanoma Treated in theIndividual Patient Program (DESCRIBE Italy) by Massimo Aglietta et al in Target Oncol 2021 Nov;16(6):789-799. doi: 10.1007/s11523-021-00850-1.

We thank the reviewer to point it out; we would like to explain better the kind of articles they are.

 The cited Aglietta et al. was a retrospective study based on data collected by the centres and resulting from their  patient’ archives. Data came from Italian centres, in a context of the compassionate use of the combination, with a cut off at 2018.

The data presented in the current manuscript concern an observational, prospective study, where clinical development of patient condition was followed, allowing, in our opinion, further insight into the use of the combination in the patient setting described.

This study is a prospective study with limitation as described by the authors “Due to the observational nature of the study, data were non-standardized since they were collected within the routine medical care framework where the medical practice may vary.”

The other major limitation was : “this study was terminated early, at a time when many patients were still being treated with the study drug, the full pattern of disease progression for these patients could not be assessed.”

We agree with the reviewer, observational studies have always some limitations,  and this was the reasons because we underline our ones, as it is usually done for any publication.

My opinion is that it is very hard to follow the results as described.

The clinical parameters of patients are missing.

We apologize, but we have not understood which kind of clinical parameters the reviewer refers to,  since we reported the characteristics of the population in Table 1.

The descriptions of supplementary Figures are superficial and the titles of Figures are superficial.

We improved the titles and descriptions.

Other notes: Simple summary: It is not clear what are the new observation of this study.

Thanks to point it out; a sentence to better explain the aim of the study were added in the summary

This manuscript is a resubmission of an earlier submission. The following is a list of the peer review reports and author responses from that submission.

Round 1

Reviewer 1 Report

This was a prospective, observational study of the patterns of first-line treatment progression with dabrafenib plus trametinib combination in BRAF V600E/K or other BRAF activating mutation-positive cutaneous melanoma patients with either limited or bulky disease in clinical practice. It is  a confirmatory, albeit useful, study. There are few points that need clarifications.

      The authors combined patients with Stage IIIC (unresectable) or Stage IV (metastatic) melanoma but stratified them based on . This is problematic in trying to understand data in Fig.1. For example, what proportion of patients with Stage IV disease still have skin lesions at baseline? The authors may want to analyze the data by stratifying patients based on disease stage.

      In Lines 108-109- the authors state that inclusion criteria included that patient “be naïve to treatment for advanced/metastatic disease without having received prior systemic anti-cancer treatment”. This is in conflict with statement in Lines 183-187 “5.8% of patients in Cohort A and 8.2% of patients in Cohort B had prior antineoplastic radiotherapy, ......” . This must be clarified/corrected. Also, information on antineoplastic regimens  administered to these patients should be mentioned.

      Fig 1a and 1d are not really informative and appear redundant with data in Figs 1b and 1e. Similarly, although data in Fig 1c and 1e are useful, a better representation could be  proportions of patients with total number (n) of metastatic lesions in all organs.

Author Response

Reviewer 1

The authors combined patients with Stage IIIC (unresectable) or Stage IV (metastatic) melanoma but stratified them based on. This is problematic in trying to understand data in Fig.1. For example, what proportion of patients with Stage IV disease still have skin lesions at baseline? The authors may want to analyze the data by stratifying patients based on disease stage.

Thank you for the comment; however, the efficacy analysis after stratifying for stage III and IV was not done.

In Lines 108-109- the authors state that inclusion criteria included hat patient “be naïve to treatment for advanced/metastatic disease without having received prior systemic anti-cancer treatment”. This is in conflict with statement in Lines 183-187“5.8% of patients in Cohort A and 8.2% of patients in Cohort B had prior antineoplastic radiotherapy, ......” . This must be clarified/corrected. Also, information on antineoplastic regimens administered to these patients should be mentioned.

We clarified the sentences as required both in Material and Methods section and in Results section.

Fig 1a and 1d are not really informative and appear redundant with data in Figs 1b and 1e. Similarly, although data in Fig 1cand 1e are useful, a better representation could be proportions of patients with total number (n) of metastatic lesions in all organs.

We added some details in the captation to better clarify the figure

Reviewer 2 Report

This is an ambitious observational study investigating the pattern of disease progression in Italian patients with unresectable cutaneous BRAF mutated melanoma treated with dabrafenib in combination with trametinib. Altough the authors using a prosepctive study design the collection of data has been done with large numbers of missing data. Potentially it seems as data was retrieved retrospectively from medical records and not from a dedicated study database with a continued collection explaining that a large proportion of data has not been captured. Correct? Please explain.

Today there is a general agreement that BRAF-mutated melanoma with low disease burden and without symptoms should preferably be offered immunecheckpoint blockade as 1L treatment and not as in this study with BRAF-MEK inhibitors. Thus, the clinical data from cohort A to some extent may be outdated but cohort B is still very relevant. Please comment on todays situation compared to how it was when the present study was initiated.

Specific comments

1.       During what exact time period were the patients enrolled in the present study. This is not given in text. Please clarify.

2.       To some extent ECOG status reflect symptomatic vs asymptomatic disease. Please comment in relation to baseline characteristics, what proportion of patients in cohort A and B respectively, had symptoms of disease?

3.       Since 2018 patients with stage III melanoma have been offered adjuvant therapy. Its not clear to me what proportions of patients had received systemic adjuvant therapy. Please clarify the number of patients and what type of medication. In the Figure 1  its said that systemic therapy was not allowed but in Table 1 there are patients given ” adjuvant antineoplastic medication”. Please clarify.

4.       Table 1 is key in understanding of the patient population. No data is given on primary tumor, TNM, stage I/II, Breslow/ulceration/type of BRAF-mutation. Proportion of different M1 status A/B/C/D is essential. Please add.

5.       What proportion of patients had unresectabkle stage III vs IV. How did this impact in terms of efficacy?

6.       The data on patients with brain metatstastasis is potentially the most interesting in the study but somewhat neglected. I suggest Figure 3 with efficacy data for patients with brain metastasis to be included in the original ms, not in the suppl. material.

7.       Collecting of response/PFS data is based on radiologiocal analysis but not described. Please add this information.

8.       Safety, how was heart function (LVEF) assessed during the study. Dis you observe any impact on LVEF? Any patients needed to quit MEK-i in the study for this reason or other reasons and thus given monotherapy?

Author Response

On behalf of my co-authors, I wish to thank you for the time spent in revising our paper and for the comments that will contribute to improve our work. 

We have considered your comments and revised the manuscript accordingly. Please, find enclosed our responses point-by-point. We trust that the manuscript has been suitably revised to address the issues raised during review.

  1. During what exact time period were the patients enrolled in the present study. This is not given in text. Please clarify.

We added this information at the beginning of the Result section.

  1. To some extent ECOG status reflect symptomatic vs asymptomatic disease. Please comment in relation to baseline characteristics, what proportion of patients in cohort A and B respectively, had symptoms of disease?

The vast majority of patients of cohort A had ECOG PS 0-1 and, presumably, were asymptomatic; in the cohort B the proportion of patients with higher ECOG increased, reflecting the higher burden of the disease.

  1. Since 2018 patients with stage III melanoma have been offered adjuvant therapy. It’s not clear to me what proportions of patients had received systemic adjuvant therapy. Please clarify the number of patients and what type of medication. In the Figure 1 its said that systemic therapy was not allowed but in Table 1 there are patients given ” adjuvant antineoplastic medication”. Please clarify.

We added this information to the text. In Italy dabrafenib and trametinib were approved for the adjuvant setting in December 2019.

  1. Table 1 is key in understanding of the patient population. No data is given on primary tumor, TNM, stage I/II,Breslow/ulceration/type of BRAF-mutation. Proportion of different M1 status A/B/C/D is essential. Please add.

We did not collect this information.

  1. What proportion of patients had unresectable stage III vs IV. How did this impact in terms of efficacy?

The impact of the initial stage on the efficacy was not evaluated.

  1. The data on patients with brain metastasis is potentially the most interesting in the study but somewhat neglected. I suggest Figure 3 with efficacy data for patients with brain metastasis to be included in the original ms, not in the suppl. Material.

Thank you for this suggestion, we added the figure 3 with efficacy data for patients with brain metastasis.

  1. Collecting of response/PFS data is based on radiological analysis but not described. Please add this information.

As this was an osservational study, radiological response and disease progression were evaluated by the treating physician as per clinical practice (please refer to lines 144-145).

  1. Safety, how was heart function (LVEF) assessed during the study. Dis you observe any impact on LVEF? Any patients needed to quit MEK-i in the study for this reason or other reasons and thus given monotherapy?

We did not collect this information.

Reviewer 3 Report

The aim of the study was to understand whether treatment of the combination of dabrafenib and trametinib will modify the progression pattern of BRAF mutated melanomas.

The authors already reported an investigation entitled: Retrospective Chart Review of Dabrafenib Plus Trametinib in Patients with Metastatic BRAF V600-Mutant Melanoma Treated in the Individual Patient Program (DESCRIBE Italy) by Massimo Aglietta et al in Target Oncol 2021 Nov;16(6):789-799. doi: 10.1007/s11523-021-00850-1.

This study is a prospective study with limitation as described by the authors „Due to the observational nature of the study, data were non-standardized since they were collected within the routine medical care framework where the medical practice may vary.”

The other major limitation was : „this study was terminated early, at a time when many patients were still being treated with the study drug, the full pattern of disease progression for these patients could not be assessed.”

My opinion is that it is very hard to follow the results as described. The clinical parameters of patients are missing. The descriptions of supplementary Figures are superficial and the titles of Figures are superficial.

Other notes: Simple summary: It is not clear what are the new observation of this study.

Abstract:

line 41: „In both cohorts, involved organs and metastases at each location decreased.” Is it correctly In both cohorts, involved THE NUMBER OF organs and metastases at each location decreased.

LINE 45: „No new safety signals were reported.”: it is not clear what this means, I recommend to described shortly.

Introduction: Lines 54-53: The numbers are not written consistently

Results

Figure 1. Is it correct that there are inconsistency between Cohort A and B and site of progression. As far as I understand correctly Cohort A has a better prognosis, despite this the % of nodal metastasis is present only in Cohort A. I can see similar inconsistency between the number of organs involved between the two groups, or the description of Figure 1 is incomplete.

References :

Some important references are missing:

Kim CG, Kim M, Hwang J, Kim ST, Jung M, Kim KH, Kim KH, Chang JS, Koom WS, Roh MR, Chung KY, Kim TM, Kim SK, Lee J, Shin SJ. First-line pembrolizumab versus dabrafenib/trametinib treatment for BRAF V600-mutant advanced melanoma. J Am Acad Dermatol. 2022 Nov;87(5):989-996. doi: 10.1016/j.jaad.2022.07.057. Epub 2022 Sep 6. PMID: 36068115

Orlova KV, et al. Real-World Experience with Targeted Therapy in BRAF Mutant Advanced Melanoma Patients: Results from a Multicenter Retrospective Observational Study Advanced Melanoma in Russia (Experience) (ADMIRE).

Cancers (Basel). 2021 May 21;13(11):2529. doi: 10.3390/cancers13112529.

Author Response

On behalf of my co-authors, I wish to thank you for the time spent in revising our paper and for the comments that will contribute to improve our work. 

We have considered your comments fand revised the manuscript accordingly. Please, find enclosed our responses point-by-point. We trust that the manuscript has been suitably revised to address the issues raised during review.

Line 41: In both cohorts, involved organs and metastases at each location decreased. ” Is it correctly In both cohorts, involved THE NUMBER OF organs and metastases at each location decreased.

Thank you for the suggestion, we modified the sentence.

LINE 45: No new safety signals were reported. It is not clear what this means, I recommend to described shortly.

We better described the safety analysis in the results section. Due to word limitation, we cannot add further details in the abstract.

Introduction: Lines 54-53: The numbers are not written consistently

We modified the numbers

Figure 1. Is it correct that there is inconsistency between Cohort A and B and site of progression. As far as I understand correctly Cohort A has a better prognosis, despite this the % of nodal metastasis is present only in Cohort A. I can see similar inconsistency between the number of organs involved between the two groups, or the description of Figure 1 is incomplete.

It is true that by looking at figures a, b, d, and e it would appear that cohort A had a worse prognosis, but for these patients it is necessary to evaluate figures c and f since the greatest impact on prognosis is due to the disease burden and the number of organs involved. We added some details to better explain the figure.

References :

Some important references are missing:

Kim CG, Kim M, Hwang J, Kim ST, Jung M, Kim KH, Kim KH, Chang JS, Koom WS, Roh MR, Chung KY, Kim TM, Kim SK, LeeJ, Shin SJ. First-line pembrolizumab versus dabrafenib/trametinib treatment for BRAF V600-mutantadvanced melanoma. J Am Acad Dermatol. 2022 Nov;87(5):989-996. doi: 10.1016/j.jaad.2022.07.057. Epub 2022 Sep 6. PMID:36068115

Orlova KV, et al. Real-World Experience with Targeted Therapyin BRAF Mutant Advanced Melanoma Patients: Results from a Multicenter Retrospective Observational Study Advanced Melanoma in Russia (Experience) (ADMIRE).

We added the paper by Orlova et al. Thank you for this suggestion.